# The Warburg Effect 97 Years after Its Discovery

**DOI:** 10.3390/cancers12102819

**Published:** 2020-09-30

**Authors:** Rosa Maria Pascale, Diego Francesco Calvisi, Maria Maddalena Simile, Claudio Francesco Feo, Francesco Feo

**Affiliations:** 1Department of Medical, Surgery and Experimental Sciences, Division of Experimental Pathology and Oncology, University of Sassari, 07100 Sassari, Italy; calvisid@uniss.it (D.F.C.); simile@uniss.it (M.M.S.); feo@uniss.it (F.F.); 2Department of Clinical, Surgery and Experimental Sciences, Division of Surgery, University of Sassari, 07100 Sassari, Italy; cffeo@uniss.it

**Keywords:** Warburg effect, oxidative metabolism, glycolysis, oncogenes, tumor therapy

## Abstract

**Simple Summary:**

The deregulation of the oxidative metabolism in cancer cells, characterized by an increased ratio between glycolysis and oxygen consumption (Warburgv effect) largely depends on metabolic and molecular variations including modifications of oxidative metabolism, activation of oncogenes that promote glycolysis and decrease oxygen consumption, defective cancer mitochondria; attempts to correct the Warburg effect represent new approaches to tumor therapy.

**Abstract:**

The deregulation of the oxidative metabolism in cancer, as shown by the increased aerobic glycolysis and impaired oxidative phosphorylation (Warburg effect), is coordinated by genetic changes leading to the activation of oncogenes and the loss of oncosuppressor genes. The understanding of the metabolic deregulation of cancer cells is necessary to prevent and cure cancer. In this review, we illustrate and comment the principal metabolic and molecular variations of cancer cells, involved in their anomalous behavior, that include modifications of oxidative metabolism, the activation of oncogenes that promote glycolysis and a decrease of oxygen consumption in cancer cells, the genetic susceptibility to cancer, the molecular correlations involved in the metabolic deregulation in cancer, the defective cancer mitochondria, the relationships between the Warburg effect and tumor therapy, and recent studies that reevaluate the Warburg effect. Taken together, these observations indicate that the Warburg effect is an epiphenomenon of the transformation process essential for the development of malignancy.

## 1. Introduction

Following the pioneering observation by Otto Warburg that an elevated glucose consumption by cancer cells is associated with a restraint of oxygen consumption and the production of lactic acid in aerobiosis (Warburg effect) [1,2], numerous efforts were dedicated to explain the genesis of the glycolytic metabolism of tumors as well as its implication in the malignant transformation. The analysis of a spectrum of hepatocellular carcinoma (HCC), including well, moderately, and poorly differentiated tumors, showed that aerobic glycolysis increases with the degree of malignancy [3,4], suggesting that the Warburg effect is correlated to tumor progression. 

The role of Warburg effect in the pathogenesis of tumors has been the object of debate in recent years [3,4]. However, recent acquisitions on cancer biochemistry and biology contributed to clarify the pathogenesis of the Warburg effect and its role in tumor progression, and suggested new approaches to HCC therapy. 

## 2. The Deregulation of the Oxidative Metabolism in Cancer

One of the first observations on the regulation of the glycolytic metabolism of cancer cells was the demonstration by Crabtree that glucose suppresses the respiration and the oxidative phosphorylation of cancer cells (the Crabtree effect) [5]. The determination of the respiratory control and oxidative phosphorylation of the mitochondria isolated from HCC ascites AH130 (Figure 1A) demonstrates an elevated acceptor control ratio (ACR), in the presence of adenosine diphosphate (ADP), in these mitochondria [6]. Chance and Hess observed, by polarography, that the respiratory inhibition, and consequent suppression of oxidative phosphorylation, induced by glucose addition to tumor cells (Crabtree effect), was preceded by a short period of respiratory stimulation (state metabolic 3; Figure 1B) [6,7]. This phenomenon (Chance–Hess effect) was explained by variations of the availability of ADP, the respiratory stimulation being induced by the ADP produced during the first phase of glycolysis (synthesis of gluose-6-phosphate -G6P- and fructose biphosphate -FBP), followed by a decrease of the respiratory rate (state metabolic 4) when ADP is transformed to adenosine triphosphate (ATP) [7]. The mitochondrial oxidative phosphorylation proceeds to pyruvate oxidation that involves the phosphorylation of ADP to ATP, but in the glycolysis 2dP-GLY and PEP are utilized as substrates and ADP is phosphorylated to ATP [7]. 

In normal cells almost all ATP is produced by the mitochondrial oxidative activity and used for ATP-dependent reactions, while in tumor cells the production of ATP by oxidative phosphorylation is low and the phosphorylation of the glycolytic glucose is elevated. Cancer cells have a relatively small number of small mitochondria with different dysfunctions, resulting in a decrease of net oxygen consumption capacity [8,9,10,11] (see later). Furthermore, it was demonstrated that mutations of mitochondrial DNA (mtDNA) represent advantages for tumor growth in nude mice [12] and mtDNA-depleted human SK-Hep1 hepatoma cells become resistant to reactive oxygen species (ROS) [13]. The fall in mtDNA, induced by inhibitors or deletion, activates Protein kinase B (AKT) signaling by causing Phosphatase and tensin homolog (PTEN) inactivation [14]. Various alterations of mtDNA in cancer cells also occur in pre-cancerous lesions [15,16,17,18] and contribute to tumor progression [19]. Interestingly, mtDNA depletion is associated with the development of multidrug resistance (MDR) [1,20,21,22] and mutations or dysfunctions of human mtDNA intensify the metastatic potential of human lung cancer A549 cells [23]. In addition to the mtDNA mutations, a decrease in the number of mitochondria contributes to the Warburg effect. The Morris hepatoma 3924A, for instance, exhibiting a pronounced Warburg effect, has a low number of functional mitochondria, capable to produce ATP [8].

The analysis of the relationships between respiration and glycolysis in neoplastic cells clearly indicates the existence in these cells of deregulation mechanisms, in aerobiosis, such as to make the mitochondria unable to oxidize the pyruvate and to justify, therefore, the increase in lactic acid production. Thus, hexokinase 2 (HK2), the major isoform expressed in tumors, plays a key role in the voltage-dependent anion channel Warburg effect [24]. It is associated with the mitochondrial outer membrane [25] where it is bound to voltage-dependent anion channel (VDAC) (Figure 2). This enables a rise of glucose metabolism [25] and inhibition of apoptosis [24] by allowing an easier contact of HK to ATP synthesized in the inner mitochondrial membrane. 

Most of the enzymatic reactions of glycolysis are reversible, but the reactions leading to the phosphorylation of glucose and F6P, with the production of G6P and FBP, catalyzed by HK and phosphofructokinase (PFK), respectively, and the dephosphorylation of PEP to pyruvate, are irreversible: they cannot be used for both glycolysis and glycogen synthesis and provide the equilibrium between cytoplasmic ADP and ATP (Figure 3). The increase of ADP concentration, produced during the glucose utilization, markedly accelerates the oxygen utilization (state 3 respiration). This is followed by a decrease in cytoplasmic ADP and an increase in mitochondrial ATP (state 4 respiration) characteristic of the Crabtree effect (Figure 1). As a consequence, a decrease of the pyruvate oxidation occurs in the mitochondrial matrix, which may also contribute to the decrease of pyruvate transport, by a specific carrier, into the mitochondrial matrix [26], the increase in pyruvate dehydrogenase kinase 1 (PDK-1) activity [27,28] and, at least in HCC, the overexpression of pyruvic dehydrogenase E1α (PDH-E1α) [29]. Interestingly, the protein levels of (β-F1-adenosine-triphosphatase (β-F1-ATPase; the physiological inhibitor of the H^+^-ATP synthase), which are reduced in human liver, kidney, colon, breast, and stomach carcinomas, was found to be correlated, in colon cancer, with both the time of cancer recurrence and patients survival [30], suggesting that the alteration of the bioenergetic function of mitochondria is a main feature of these cancer types. 

Other mechanisms may contribute to the Crabtree effect: the HK2 bound to mitochondria [31], the rise in FBP concentration, produced by the high glycolytic flux, which may inhibit the complexes 3 and 4 at the level of the mitochondrial respiratory chain [32], and the active glycolytic flux that may compete with mitochondria at level of the Pi carrier and adenine-nucleotide transport (Figure 2). The modulation of the transformation of pyruvate to acetyl-CoA, by pyruvate dehydrogenase (PDH), may also contribute to the Crabtree effect (Figure 2). Indeed, the PDH inactivation by phosphorylation, catalyzed by a specific kinase, and its reactivation by PDH phosphatases [27] may regulate the supply of acetyl-CoA to Krebs cycle. The inactivation of PDK-1 was shown to decrease the lactate production in the head and neck squamous cancer cells [27]. Indeed, PDK-1 inhibits PDH [33] and consequently suppresses the transformation of pyruvate into acetyl-CoA, the Krebs cycle, and the reactions leading to the resynthesis of PEP and G6P.

## 3. The Genes Involved

Many oncogenes promote glycolysis and are involved in the decrease of oxygen consumption in cancer cells (Figure 3). *H-ras* activates phosphoglycerate kinase (PGK), enolase and PK (pyruvate kInase) [34] and *Myc* induces the splicing factors involved in the production of phosphoglycerate kinase M2 (PKM2) [35], an isoenzyme that promotes glycolysis in aerobiosis. *Myc* also activates the expression of LDH [36,37,38], glutaminolysis [37,39], HK2 [37,38,39,40], and I [36,37]. PFK2 is also activated by the *LKB1*/*AMPK* (Liver Kinase B1/Liver Kinase B1 axis [38]. 

The PI3K/AKT pathway stimulates lipid metabolism and protein synthesis, and contributes to glycolysis by the activation of numerous enzymes (Figure 3). The process involves the binding of AKT to the cell membrane, with the help of the phosphoinositide-dependent kinase. The PI3K/AKT pathway activates the transport of glucose, as well as HK and PFK [41,42], decreases glycogen synthesis by inhibiting the glycogen synthase kinase 3β via phosphorylation of its N-terminal serine [42]. This leads to the accumulation of cyclin D1, that promotes the cell cycle progression and contributes to the mutation of tumor suppressor genes, such as *p53* [42]. AKT signaling also inhibits apoptosis [42] and controls 6-phosphofructo-2-kinase/fructose-2,6-biphosphatase 3 (PFKFB3) activity [43]. The activation of PFK by *AKT* also abolish the inhibition of (phosphofructokinase 2) PFK2 by ATP [44]. 

HIF1α (Hypoxia-inducible factor 1-alpha), is mainly involved in the glycolysis in anaerobiosis [45,46,47] (Figure 3). It is frequently overexpressed in cancer cells [48,49]. In the hypoxic conditions of these cells *HIF1α* is stabilized and translocates into the nucleus [50,51]. HIF1α mediates the expression of PDH-K1 that phosphorylates PDH and inhibits its activity, thus contributing to the down-regulation of mitochondrial respiration [51,52]. The stimulation of HIF1α expression in aerobiosis, consequent to the low level of oxygenation of the neoplastic tissue, indicates that HIF1α also plays a role in glycolysis in aerobic conditions by stimulating the transport of glucose, HK, PFK, aldolase, enolase, LDH, PKM2, PDH-K1 (pyruvate dehydrogenase K1) and MCT4 (monocarboxylate transporter 4. Figure 3) [50,52]. The latter protein, frequently overexpressed in cancer cells, facilitates the translocation of pyruvate and lactate through the plasma membrane, with consequent acidification of the extracellular matrix [53,54]. The role of HIF1α is also shown by its activation of glycolysis, induction of *MYC* and *RAS* overexpression and loss of p53. These effects are mediated by a family of regulatory bifunctional PFKFB proteins [54,55,56,57,58]. Also, HIF1α regulates the cytochrome oxidase isoform 4-2 and LON, a mitochondrial protease that is required for COX4-1 (Cytochrome c oxidase subunit 4 isoform 1), which degrades cytochrome oxidase [57]. Furthermore, it induces the protein BNIP3 (BCL2/Adenovirus E1B 19-KD protein-interacting protein 3) that, under persistent hypoxia, primes cells for autophagy [59].

The mitochondrial pyruvate carrier complex (MPC), of the inner mitochondrial membrane, transports the pyruvate to mitochondrial matrix. Therefore, the MPC complex is a regulator of glycolysis in tumor cells as, under hypoxic conditions, lactate secretion from cancer cells increases, while MPC1 and MPC2 levels decrease [60]. The human HK2, overexpressed in all aggressive tumors, is predominantly located in the outer mitochondrial membrane, where the interaction through its N-terminus increases its stability and maintains tumorigenesis [26,61]. The predominant role of HK2 in tumor cells is also confirmed by the observation that the tumor subgroups expressing both HK1 and HK2 are sensitive to inhibition of HK2 alone [62].

Finally, the adaptive response to hypoxia in cancer cells contributes, through the overexpression of *HIF-1α,* to the activation of glucose transport and thus to glycolysis and the pentose phosphate pathway [53]. The glycolytic metabolism of cancer cells has been related to a broad spectrum of mutations and depletions present in human cancers. Thus, the activation of oncogenes and mutations of oncosuppressor genes, including p53, have been considered responsible for the upregulation of glycolytic enzymes and the inhibition of the biogenesis or assembly of respiratory enzyme complexes, such as cytochrome c oxidase [63]. The lactate synthesis from pyruvate regenerates the NAD+ necessary to complete the glycolysis (Figure 3). 

The export of lactate outside of the cell, through MCT1–4 which are upregulated in different cancer types [64], has different effects (Figure 4). Indeed lactate enters tumor endothelial cells and stimulates VEGF protein expression, angiogenesis and tumor growth [65,66,67]. Furthermore, the interstitial acidification provides a protection from the immune system since elevated lactic acid concentrations impair the activity of cytotoxic T lymphocytes and leucocytes to kill cancer cells [68]. Finally, the inhibition of LDH arrests cancer cell growth, progression, and invasivity [69,70,71]. Interestingly, MCT1–4 are a target of *TIGAR*, a gene overexpressed in cancer that is associated with chemotherapy resistance and could be considered a novel therapeutic target [72].

An important role in the metabolic reprogramming of cancer cells is played by the TGF-β1 (Transforming growth factor beta 1) through the upregulation of Smad, p38 MAPK and PI3K/AKT pathways [73]. Specifically, TGF-β1 promotes tumor progression by inducing a switch from an epithelial to a mesenchymal/migratory phenotype in HCC cells, reducing mitochondrial respiration and enhancing glutamine transporter SLC7A5b (Solute Carrier Family 7 Member 5), glutaminase 1, and pentose phosphate cycle [74]. The latter provides precursors for nucleotide synthesis and prevents oxidative stress and redox homeostasis. Furthermore, TGF-β1 and its mediated signaling pathway induces HIF1α to participate in the process of metabolic reprogramming under normoxic conditions [74].

Most of the aforementioned metabolic deviations in cancer cells, including the central role of HIF1α, have been confirmed by proteomic researches [49,75,76,77]. Thus, it has been found that HIF-1α induces the expression of PDK-1 that phosphorylates and inactivates PDH [77]. The consequent decrease in acetyl-CoA induces that of carboxylic acid cycle. Proteomic studies of early stages of breast and prostate cancer confirmed the pivotal role of HIF1α as a transcription factor [78]. HIF1α expression was found to be correlated with prognostic indicators for metastatic disease and considered a biomarker potential prognostic of breast, prostate, lung and gastric cancers [79].

Proteomic studies [75,80,81] also underlined the role of the cellular signaling pathways regulated by PI3K/AKT kinases in induction of the Warburg effect [81,82]. In particular, proteomic experiments confirmed the prognostic role of some enzymes and/or isoenzymes of the glycolytic pathway such as HK2 [24,25], GPI (gluose-6-phosphate isomerase) [24,26] and LDH [83]. Those studies also showed the possibility of reducing the hypoxia-linked cancer cell resistance by different mechanisms such as: (a) hampering the HIFα-linked survival pathways; (b) inducing a cytotoxic synergism with conventional voltage-dependent anion channel cancer treatments; (c) improving cancer drug selectivity; and (d) deranging ATP-dependent multidrug resistance [75].

## 4. The Genetic Susceptibility to Cancer

Previous study of the genetic mechanisms of the inherited predisposition to HCC, based on the comparative evaluation of the molecular pathways involved in HCC developing in rat strains differently predisposed to HCC, have shown that the genes responsible for the susceptibility to HCC control the amplification and/or overexpression of c-*Myc*, the expression of cell cycle regulatory genes, the activity of *Ras/Erk* and *Akt/mTor*, the methionine cycle, and DNA repair pathways [84]. The activation of oncogenes or the mutation of oncosuppressor genes, including p53, may be responsible of different deregulations leading to glycolytic metabolism and to the inhibition of the respiratory enzymes including cytochrome c oxidase. Previous work in our laboratory [33,85,86] demonstrated the overexpression of c-*Myc, Cyclins D1, E*, and *A*, and *E2f1* genes, at mRNA and protein levels, and the increase of CDK4 (Cyclin-dependent kinase 4), Cyclin E-CDK2 (Cyclin-dependent kinase) and E2f1-DP1 complexes, and pRb hyperphosphorylation in dysplastic nodules and HCCs of the F344 rats, genetically susceptible to hepatocarcinogenesis. These changes were absent in the resistant BN rat lesions, in which Dusp1 (Dual Specificity Phosphatase 1) caused low *Erk* activation, whereas a Dusp1 decline, associated with high *Erk* activation, was found in the susceptible F344 (Fischer 344) rat lesions [86]. Interestingly, Dusp1 is slightly upregulated in preneoplastic liver of both F344 and BN (Brown Norway) strains, but its level decreases in early dysplastic nodules and HCCs of F344 rats, whereas it further increases in the lesions of BN rats [86]. This suggests that even in the presence of a limited increase in the levels of upstream activators of *Erk1/2* (*Raf1*, *Mek1/2*), a decrease of Dusp1 sustains *Erk1/2* activation and contributes to the development of autonomously growing liver lesions of F344 rats. Furthermore, we observed that c-*myc* overexpression and amplification were correlated with the propensity of dysplastic nodules to progress to HCC in poorly susceptible Wistar rats [87]. These researches demonstrated that the genetic alterations of preneoplastic and neoplastic lesions of resistant rats correspond to those of human HCC with better prognosis, while the alterations of the lesions of susceptible rats correspond to those of human HCC with poorer prognosis. Accordingly, a comparative functional genomic analysis allowed for the discovery of an evolutionarily conserved gene expression signature discriminating HCC with different propensity to progression in rat and human, indicating that the molecular pathways associated with specific cancer phenotypes are evolutionarily conserved [88]. The gene signature differentiating rat and human cancer subtypes differently prone to progression included the genes *BHMT* (Betaine-Homocysteine S-Methyltransferase) and *GNMT* (glycine N-methyltransferase), involved in the maintenance of the hepatocyte SAM (S-adenosylmethionine) level and, hence, in the regulation of the susceptibility to cancer progression [88]. These observations indicate that the same mechanisms regulate the growth and progression of preneoplastic and neoplastic liver lesions in susceptible and resistant rats. Accordingly, polarographic analysis has shown that analogous mechanisms regulate the respiratory rate and the oxidative phosphorylation of well-coupled mitochondria isolated from the highly undifferentiated AH-130 HCC and the moderately-differentiated 5123 HCC [89]. Furthermore, we found that SAM, a compound with an anticarcinogenic effect [90], inhibits enolase, FBPase (fructose biphosphatase), ME (malic enzyme), PK, and G6PDH (glucose-6-phosphate dehydrogenase) in preneoplastic liver lesions, thus inducing a partial reversion of the carbohydrate metabolic features of these lesions to those present in normal liver [91].

These findings are in keeping with the Weber’s “molecular correlation” concept [92] on the basis of which key reactions of glycolysis are positively or negatively correlated to the degree of biologic deviation of tumor tissue from normal tissue. Therefore, the increasing lactic acid production in aerobiosis in liver tumors with different growth rate (Figure 3) merely reflects the link between the Warburg effect and tumor progression [93,94]. 

## 5. The Molecular Correlations

The Warburg pioneering observations on cancer metabolism encouraged several researches dedicated to understanding the molecular changes responsible for cancer transformation. It was thus observed that the ATP necessary for the biosynthesis of the compounds needed for cancer cell proliferation is provided by glycolysis, a process that makes ATP available to cells with less efficiency, but more rapidly than respiration [95]. It was calculated that as a consequence of the Warburg effect the glycolytic flux is so enhanced in cancer cells that by the time that one ATP molecule is produced by respiration, the aerobic glycolysis generates 24 ATP molecules [96]. In a stimulating review, Sciacovelli and coworkers [97] underlined the metabolic reprogramming leading to enhanced glycolysis, alterations of mitochondrial function and lipid metabolism involved in cancer transformation. Among the enzymes involved, PFK1 (phosphofructokinase muscle type) undergoes glycosylation that inhibits its activity and redirects glucose through the pentose phosphate pathway [98]. Also, ATP in cancer cells is used to maintain the electrochemical gradients and protein turnover, more than for the generation of biosynthetic precursors required for cell division [99].

PKM2 expression and localization exerts an important role in cancer growth and survival [100]. The regulation of PKM2 is complex. When PKM2 is inactive, glycolytic components preceding PEP tend to accumulate and are switched to the pentose phosphate pathway [101] and to the biosynthesis of serine [102]. Under hypoxic conditions, HIF1α binds to the promoter region of PKM2 [103], thus promoting the transactivation of HIF1α downstream genes [104], including PKM2 [105].

PKM2 may be present in the cells as inactive dimers and active tetramers (Figure 5). Dimeric PKM2 is activated to the tetrameric form by FBP while it is inactivated by a high ATP concentration [106]. Phosphorylated PKM2 tetramers induce the transformation of pyruvate to OAA utilized in Krebs cycle for oxidative phosphorylation. PKM2 dimers phosphorylated, for instance by active ERK [107], enter the nucleus where they favor the biosynthesis of nucleotides, phospholipids and amino acids [108]. They can also regulate *myc* expression [107,109], including *HK*, *PFK*, and *LDH* genes [107] (Figure 5). Active PKM2 tetramers interact with the dioxygenase/demethylase, JMJD5 (Jumonji domain containing 5) [110] (Figure 5). In the nucleus, the complex JMJD5/PKM2 modulates the *HIF1α*-induced expression of metabolic genes [111]. Dimeric PKM2 acetylated in the presence of serine, enters the nucleus where it interacts with Oct-4, to enhance Oct-4-mediated transcription potential [112], and activates STAT3 by phosphorylating its Tyr305 residue, PEP being the phosphate donor [113,114]. This activates the transcription of genes involved in cell proliferation, such as *mek5* [115] and *Hif1α* [104,116] (Figure 5). Tumor cell proliferation is also supported by PKM2-dependent phosphorylation the Thr11 of histone H3, favoring its acetylation, and the removal of HDAC3 (histone deacetylase 3) from *CYCLIN D1* and *MYC* promoters [117]. 

These findings reveal the multifunctional role of PKM2 in tumors. A positive feedback loop promotes HIF1α activity and maintains high expression of *PKM2* and other glycolytic genes. However, in apparent contrast with these findings, it was observed that PKM2 deletion accelerates tumor formation in a BRCA1-deficient breast cancer mouse model, suggesting that PKM2 is not necessary for tumor cell proliferation [118]. Furthermore, it was found that the highly proliferative PKM2-deficient cancer cells also have a low expression of PKM1, suggesting a negative correlation between PK activity and proliferation in these tumor cells [118]. These findings, however, contrast with the above-mentioned evidence of a PKM2 role in carcinogenesis and they could represent the specific behavior of the BRCA1-deficient mouse breast cancer cells. Finally, PKM2′s nonmetabolic functions, as a transcriptional coactivator and protein kinase (particularly as a kinase regulating histone phosphorylation and acetylation), provide PKM2 with the capacity to regulate gene expression, cell cycle progression, and metabolism [119].

A possible oncogenic role has also been attributed to PHGDH (phosphoglycerate dehydrogenase), involved in the synthesis of serine, pyruvate, and hydroxypyruvate, in breast and skin cancers and glioma [120,121]. It should be noted, in this respect, that 3-PG (3-phosphoglycerate) inhibits the 6-PGDH and 2-PG (2-phosphoglycerate) activates PHGDH and thus the synthesis of SAICAR, a purine synthesis intermediate that activates PKM2 and, consequently, pyruvate synthesis [122] (Figure 6).

Tp53, Sirt1 (8irtuin 1), shRNA, MJE3 and PMI-004A inhibit PGAM1 (Phosphoglycerate mutase 1) [123,124,125,126] (Figure 6). This should maintain 3-PG levels elevated while 2-PG, PHGDH activity and 6-PGDH (6-phosphoglycerate dehydrogenase) and the pentose phosphate pathway are inhibited. Nm23-H1, hypoxia, 2,3-BPG (2,3-biphosphoglycerate) and dCTP (Deoxycytidine triphosphate) are PGAM1 activators (Figure 6). They can keep the intracellular 3-PG at low levels, and the 6-PGDH inhibition by 3-PG should be abrogated. Furthermore, the main function of PGAM1 is the preservation of 2-PG physiologic levels that sustain PHGDH activity (Figure 6). As a consequence, more 3-PG is required for serine synthesis and this contributes to maintain low 3-PG levels in cancer cells [124]. The inhibition of PKM2 by FGFR1 maintains elevated levels of PEP (phosphoenolpyruvate) that activate PGAM1 by transferring the phosphate to the enzyme thus producing pyruvate (Figure 6). This mechanism supplies the absence of PKM2 activity and maintain the glycolytic flux [127]. Thus, PGAM1, overexpressed in cancer, as a consequence of the downregulation of TP53, contributes to maintain low cellular levels of 3-PG and elevated 2-PG levels. 3-PG binds to and inhibits 6-PGDH and the phosphate pathway, while 2-PG activates PHGDH thus controlling the 3-PG levels [126]. When PKM2 activity is inhibited by FGFR1, PEP can activate PGAM1 by transferring the phosphate (Figure 6). In the absence of PKM2 activity, PEP can be converted into pyruvate by catalyzing the activity of PGAM1, thus maintaining the glycolytic flux [127].

The alterations of lipid metabolism in cancer cells have been under-evaluated for long time. Previous observations demonstrated that carbons from glucose and acetate are incorporated into lipids in cancer cells [128]. Furthermore, it was demonstrated that FASN is upregulated in breast cancer patients. Citrate, synthesized in the Krebs cycle, is exported in the cytosol and is converted to acetyl-CoA and oxaloacetate by ACLY [129] (Figure 7). Acetyl-CoA is transformed to malonyl-CoA by ACC and the acetyl and malonyl groups are assembled by FASN to produce palmitic acid. It should be considered that under the partial hypoxia that characterizes tumors, citrate is preferentially synthesized from glutamine, which thus represents an important supplier of fatty acids [129]. Furthermore, the physicochemical properties of cell membranes largely depend on the degree of lipid saturation. Thus, a rise of saturated fatty acids renders cell membranes resistant to oxidative stress [130], while reducing the uptake of chemotherapeutic agents [131]. 

Another important function of glutamine is the support of HIF1-dependent hypoxic response (Figure 7). The hypoxic tumor microenvironment activates glutamine and proline metabolism via ALDH18A1 (aldehyde dehydrogenase 18A1). Furthermore, hypoxia decreases the activity PRODH2 (Hydroxyproline dehydrogenase 2) with consequent accumulation of hydroxyproline, which stabilizes HIF1α and promotes HIF-dependent cell survival and HCC resistance to sorafenib [132].

Furthermore, glutamine metabolism fuels the tricarboxylic acid (TCA) cycle, nucleotide and fatty acid biosynthesis, and redox balance in cancer cells. Glutamine activates mTOR (mammalian target of rapamycin) suppresses endoplasmic reticulum stress, and promotes protein synthesis. Glutamine enters mammalian cells through glutamine transporter SLC1A5 (Alanine, Serine, Cysteine Transporter 2) [133] and contributes to different metabolic pathways. It is converted by glutaminase to glutamate that can contribute to the synthesis of reduced glutathione (GSH) [134] or is transformed by glutamate dehydrogenase or aminotransferases [135] to αKG (α-ketoglutarate) which enters the Krebs cycle where it is converted to citrate. The latter undergoes a reductive carboxylation to produce citrate, which supports the synthesis of acetyl-CoA and lipids [136] or is transformed to aspartate that supports nucleotide synthesis [137].

## 6. Cancer Cell Mitochondria

The carbohydrate metabolism of cancer cells exhibits the above described important deviations and cancer mitochondria are well coupled (Figure 1) [7,89,138]. The upregulation of glycolysis is a hallmark of cancer representing a metabolic action for the functional needs of cancer cells [139,140,141]. Due the low activity of PDH, the Krebs cycle is principally fueled by PyrCX (pyruvate carboxylase) and glutamine (Figure 3) [142,143], in the presence of hypoxia and mitochondrial disfunction of cancer cells (see above).

Purine and glutamine syntheses use the amide nitrogen of glutamine and glutamine-derived carbons respectively. The latter is also used for amino acid and lipid synthesis. Glutaminolysis, often determined by c-*Myc* (Figure 3), characterized by glutaminase upregulation that transforms glutamine to glutamate and ammonia, is high in many tumors, [144]. Glutamate is oxidized by glutamate dehydrogenase to α-KG (α-ketoglutarate) and further processed in the Krebs cycle. Glutamine uptake is inhibited by the tumor suppressor SIRT4 (sirtuin 4. Loss of SIRT4 in Eμ-myc B cell lymphoma increases glutamine consumption and accelerates tumorigenesis [145]. In addition, transaminases utilize glutamate nitrogen to couple α-KG production to the synthesis of non-essential amino acids. 

Different oncogenes control mitochondrial biogenesis and metabolism. *c-Myc* simulates cell cycle progression and glycolysis, promotes mitochondrial respiration and generates ROS [146]. PI3K mutation or loss of the *PTEN* tumor suppressor induces mTOR activation that stimulates mitochondrial biogenesis [147]. The peroxisome proliferator PGC-1α is a regulator of mitochondrial biogenesis [148]. Chronic nutrient deprivation and low energy conditions induce a rise of AMP/ATP ratio and AMPK activation which also promote mitochondria biogenesis [149]. 

*Tp53* inhibits glycolysis and drives the transcription of genes required for the maintenance of electron transport [150]. Other *Tp53* roles in tumorigenesis are its adaptation to metabolic stress, as well as the upregulation of mitochondrial fatty acid oxidation and oxygen consumption, thus allowing cancer cells to survive in starvation conditions [151]. In addition to transcriptional regulation of mitochondrial activity, *Tp53* directly functions at the mitochondria level to induce apoptosis in response to stress via interactions with Bcl-2 family members [152]. However, frequent p53 mutations in cancer cells interrupt this interaction and do not elicit mitochondrial outer membrane permeabilization [152]. Therefore, *Tp53* mutations can promote cancer cell survival by interfering with the mitochondrial functions.

As already shown, HIF1α exerts a significant role in tumorigenesis. It stimulates glycolysis in low oxygen conditions and inhibits mitochondrial respiration [153]. ROS produced by mitochondria regulate HIF1α effects by inhibiting PHDs (prolyl hydroxylase domain enzymes), negative regulators of HIF signaling. The mitochondrial deacetylase SIRT3 (Sirtuin 3) maintains ROS and redox homeostasis by deacetylating and activating the mitochondrial SOD2 and IDH2 and upregulating the antioxidant pathways [154,155,156]. This is associated with HIF1α degradation, decrease of glycolysis, and thus the Warburg effect [156,157]. A contribution to the determination of the glycolytic metabolism of cancer cells is also given by the oncogene *RAS* through the increment of the expression of the glucose transporter GLUT1 (glucose transporter type 1) [158] and of key glycolytic enzymes such as enolase, PFK, and LDH [33,159,160] (Figure 3). 

A major downstream effector of the PI3K/AKT signaling is mTOR, which takes part in mTORC1 and mTORC2 (mechanistic target of rapamycin complex 1 and 2) signaling complexes. It is involved in the regulation of mitochondrial biogenesis and stimulates, through mTORC1, various mitochondrial metabolic pathways. Thus, the transcriptional repression of the mitochondrial enzyme SIRT4 (Sirtuin 4) by TORC1 leads to GDH (glutamate dehydrogenase) activation which upregulates glutaminolysis [161]. Furthermore, mTORC1 enhances the transcription of the mitochondrial methylenetetrahydrofolate dehydrogenase 2 enzyme involved in purine synthesis [162]. 

In HCC, frequent mitochondrial dysfunctions and point mutations in the mitochondrial DNA coding region have been documented [163]. These alterations and the decrease in the mtDNA copy number, correlated with larger tumor size, liver cirrhosis, and poor survival, provide a molecular basis for the Warburg effect [163]. In the human SK-Hep1 hepatoma cell line, mtDNA depletion induces resistance to oxidative stress and chemotherapeutic agents through a rise of antioxidant enzymes [163]. Furthermore, respiratory inhibitors and inhibitors of mtDNA replication induce cisplatin resistance in human hepatoblastoma HepG2 cells and promote cell migration in other HCC cells via a paracrine signaling pathway [13].

## 7. Defective Mitochondrial Biogenesis

The transcription coactivator peroxisome proliferator-activated receptor gamma, coactivator 1 alpha (PGC-1α) is a regulator of mitochondrial biogenesis acting as a stress sensor in cancer cells. It can be activated by nutrient deprivation, oxidative damage, and chemotherapy interacting with various transcription factors [164]. Invasive cancer cells use PGC-1α overexpression as a strategy to reinforce a switch to glycolytic metabolism in low oxygen conditions [165]. Furthermore, PGC-1α stimulates the transcription-dependent mitochondrial biogenesis [166]. These findings are relevant in vivo, as in circulating tumor cells from primary orthotopic breast tumors, showing increased mitochondrial biogenesis and respiration, PGC-1α silencing attenuates the formation of metastasis [166], indicating that PGC-1α dependent mitochondrial biogenesis may contribute to tumor metastatic potential.

## 8. Interactions between Glycolysis and Mitochondrial Oxidative Activity: The Reverse Warburg Effect

Recent work reviewed in [167] highlighted the interplay between glycolysis and oxidative activity as a relevant mechanism providing to survival for cancer cells. Indeed, cancer cells, depending on surrounding conditions, could produce some energy via mitochondrial oxidative phosphorylation (OXPHOS). The stromal cells around the tumor (especially carcinoma-associated fibroblasts or Cafs) may convert the lactate produced by tumor cells, to pyruvate, which can fuel mitochondrial OXPHOS. This phenomenon, called the ‘‘reverse Warburg effect’’ [168], indicates the existence of increased aerobic glycolysis in stromal cells adjacent to tumor cells [167]. MCT1 and MCT4 (monocarboxylate transporter 1,4), have a key role in this process. Indeed, the influx of lactate by oxidative cancer cells occurs through MCT1 whereas lactate is released through that thereby induces glucose consumption, while MCT4 inhibition can directly induce a rise in intracellular lactic acid in hypoxic tumor cells [169]. Furthermore, the lactic acid shuttle avoids the formation of a fatal acidic environment of cancer cells [170,171]. Interestingly, previous observations [172] showed that the lactic acid released through MCT4 from glycolytic tumor cells can induce angiogenesis and tumor growth through an IL-8 dependent pathway.

The rapid growth of cancer cells implies the generation of nucleotides, amino acids, lipids, and folic acid for cancer cell division. NAD (Nicotinamide adenine dinucleotide) mediates redox reactions in numerous metabolic pathways, including glycolysis [65]. Therefore, increased NAD levels enhance glycolysis and fuel cancer cells. Accordingly, Nampt (nicotinamide phosphoribosyltransferase), a rate-limiting enzyme for NAD synthesis in mammalian cells, is frequently amplified in cancer cells. NAD, a substrate for PARP (polyADP-ribose) polymerase, sirtuin, and NAD glycohydrolase (CD38 and CD157) regulates DNA repair, gene expression, and stress response. Thus, NAD metabolism is implicated in cancer pathogenesis beyond energy metabolism.

## 9. The Warburg Effect and Tumor Therapy

According to available data, cancer is one of the deadliest health problems. New therapeutic treatments should address the major factors implied in the resistance of tumors to standard treatments. Metabolic deviations can vary among different types of cancers and different patients. The tumor heterogeneity requires strategies targeting the various metabolic deviations of cancers. Nowadays, metabolic-oriented research offers different new perspectives, which may contribute to the development of innovative and effective therapeutic treatments.

### 9.1. Therapeutic Effect of the Glycolysis Inhibition

Previous work in our laboratory showed a decrease of SAM liver content during the development of preneoplastic nodules and HCC induced in rats by diethylnitrosamine. The treatment of rats with SAM reconstituted the SAM pool and inhibited the growth of nodules and HCCs [173]. We further observed [174] that the administration of SAM causes a consistent fall of the number and DNA synthesis of preneoplastic liver lesions associated to a decrease in liver PK, LDH and GPDH (Glycerol-3-phosphate dehydrogenase). SAM did not affect these enzymatic activities in normal liver, but caused a consistent decrease in initiated rats. Enolase, FBP, and ME activities also increased in the liver of initiated rats, but were not significantly affected by SAM. These results clearly showed the association of the inhibition of some glycolytic enzymes with the arrest of the growth of preneoplastic and neoplastic liver lesions. They were in accordance with the previous demonstration that the inhibitor of HK and GPI, by 2-DG [92], completely hampers the aerobic glycolysis and protein synthesis of the hepatoma ascites AH-130 but is ineffective in the rat bone marrow cells and cells isolated from chicken embryo, where the energy necessary for the protein synthesis is given by respiration. As expected, 2-DG inhibits the glycolysis and protein synthesis of all tissues tested in anaerobiosis [92]. In accordance with these findings, the inhibition of HK2 in human liver tumor Huh7 and HepG2 cells inhibits cell growth and increases cell death [175]. Furthermore, 2-DG induces apoptosis of neuroblastoma SK-N-BE (2) cells [176] and its chronic dietary administration inhibits tumor incidence in a mouse model of hepatocarcinogenesis [177]. Moreover, 2-DG synergizes with Newcastle disease virus to kill breast cancer cells to inhibit GA-3-P dehydrogenase [178], and increases the efficacy of Adriamycin and paclitaxel in human osteosarcoma and non-small cell lung cancers in vivo [179]. 

Enzo et al. observed that active glycolysis is needed for the full activity of the transcriptional cofactor YAP and TAZ (Yes-associated protein/transcriptional coactivator with PDZ-binding motif) and the primary human mammary tumors with active YAP/TAZ progress toward more advanced malignant stages [180]. YAP and TAZ are two transcriptional coactivators that promote cell proliferation through a transcriptional program intermediated by TEAD transcription factor [181]. It was also found that YAP1 is genetically controlled in rat liver cancer and determines the fate and stem-like behavior of the human disease [182]. YAP increases glycolysis [179] and supports the expression of Glut3, a known driver of a cancer stem cell phenotype, whose expression is elevated in cancer [18,182]. YAP/TAZ/TEAD and AP-1 form a complex that synergistically activates target genes controlling S-phase entry and mitosis [183]. In addition, in HepG2, Huh7, and Hep3B cells, forced YAP1 over-expression results in the expression of stem cell markers and increases cell viability [182]. This does not occur if YAP1 expression is inhibited by specific siRNA or the cells are transfected with mutant YAP1 that does not bind to TEAD [182]. 

PFK catalyzes the rate-limiting phosphorylation of F6P to F-1,6-BP, an energy consuming step of glycolysis (Figure 3). One small molecule, PFK15, inhibitor of PFAKFB3, suppresses glucose uptake and growth in Lewis lung carcinomas in syngeneic mice and has anti-tumor effects in three human xenograft models of cancer in athymic mice [184]. Furthermore, the inhibition of PFKFB3 impedes glucose uptake, glycolysis, and growth of HER2-expressing breast cancer cells [185]. Treatment with lapatinib, an FDA-approved HER2 inhibitor, decreases PFKFB3 expression and glucose metabolism in HER2+ cells [185]. In vivo administration of a PFKFB3 antagonist significantly suppresses the growth of HER2-driven breast tumors [185]. The competitive inhibitors of FKFB3, 3PO and PFK15, decrease glucose metabolism and the proliferation of different cancer cells and inhibit the growth, metastatic spread and glucose metabolism in three human xenograft models of cancer in athymic mice [186]. Finally, also GAPDH is a potential cancer therapeutic target as well, since inhibition of this glycolytic enzyme by 2-DG inhibits cancer cell growth [187].

Attempts by Shikonin to inhibit of PKM2 have shown a significant inhibition of glycolysis in tumors [188]. However, this compound is highly toxic and poorly soluble. Better results were obtained with metformin, less toxic, which increases the sensitivity to chemotherapeutic drugs by inhibiting PKM2 [189]. 

The inhibition of LDH is another proposed strategy to arrest tumor growth. Specifically it has been shown that LDH suppression induces oxidative stress and reduces tumor progression [190,191]. Its association to the inhibition of four glycolysis pathway molecules (GLUT1, HKII, PFKFB3, PDHK1) using WZB117, 3PO, 3-bromopyruvate, Dichloroacetate inhibitors (Phloretin, Quercetin, STF31Oxamic acid, NHI-1) results in an increase of extracellular glucose, a decrease of lactate production and a rise in apoptosis of cancer cells [192]. In MIA PaCa-2 human pancreatic cells, the inhibition of LDHA by GNE-140 for 2 days was found to trigger cell death, although the activation of the AMPK-mTOR-S6K signaling induced resistance to GNE-140 and increased oxidative phosphorylation [193]. Thus, the combination of the LDHA inhibitor with compounds targeting the AMPK-mTOR-S6K signaling was proposed as a potential strategy to reduce the emergence of resistance to LDHA inhibition [194]. Also, oxamate significantly suppresses the proliferation of NSCLC cells, while it exerts a much lower toxicity in normal cells [195]. Moreover, the use of miR-30a-5p, an LDHA inhibitor, has been proposed as another possible strategy to counteract the resistance to chemical inhibitors [190].

The inhibition of the lactic acid transporters MCT1/2 slows down cancer cell proliferation [196,197,198]. Mechanistic studies of MCT1 inhibitors suggest that loss of LDHA activity may generate ROS through negative feedback effects on glycolysis and the synthesis of GSH [199]. Cancer cells do not metabolize lactate but export it thus inducing the acidification of the tumor surrounding [200]. The induction of inflammation by lactate efflux attracts immune cells including macrophages. The secretion of macrophage cytokines and growth factors if not reach deadly concentrations for cancer cells, may stimulate tumor cell growth and metastasis formation [201,202]. It should be noted that MCT (monocarboxylate transporter) also functions for lactate import into cells that oxidize lactate, such as heart and skeletal muscle, or metabolize for gluconeogenesis such as liver and kidney [203]. Low MCT levels are present in many issues, but marked increases in the MCT1 and/or MCT4 levels occur in several human tumors, which thus might be treated with MCTs inhibitors.

Clinical trials with glycolytic inhibitors have also been performed [204] with 2-DG (trials I/II), the HK inhibitors Lonidamine (trials II/III) and 3-Bromopyruvate (pre-clinical), Imatinib, an inhibitor of Bcr-Abl tyrosine kinase that causes a decrease in HK and G6PD activities (approved for clinical use), and Oxythiamine, which suppresses pentose phosphate pathway by inhibiting transketolase and inhibits pyruvate dehydrogenase. The above findings clear indicate the contribution of glycolytic energy to the growth of cancer cells thus confirming the old findings [171] on the therapeutic effect of the glycolytic inhibitor 2-DG.

In a recent interesting review, Cassim and coworkers [205] report that mitochondria play a key function in tumorigenesis. To explain the role of LDH in tumor growth, these authors disrupted the *LDHA* and *LDHB* genes in the human colon adenocarcinoma and murine melanoma cells [206]. The knockout of each of these genes did not strongly reduced lactate secretion, which was instead fully suppressed in the double knockout cells (LDHA/B-DKO). Under normoxia, LDHA/B-DKO cells survived by shifting their metabolism to oxidative phosphorylation (OXPHOS), leading to a 2-fold decrease in proliferation rates both in vitro and in vivo. However, under hypoxia the LDHA/B suppression completely abolished in vitro growth, indicating its dependency on OXPHOS. These conditions were reproduced pharmacologically by treating WT cells with the LDHA/B-specific inhibitor GNE-140. These findings demonstrate that the Warburg effect is not only based on the high expression of LDHA, since both LDHA and LDHB need to be deleted to suppress fermentative glycolysis. Furthermore, on the basis of these findings, the Warburg effect does not seem to be indispensable even in aggressive tumors. Hence, according to the authors, the mitochondrial metabolism should be a target of innovative anti-cancer agents. However, the problem of the translation of preclinical drugs, effective in eradicating cancer cells, to the clinical use still remains. Indeed, these drugs would necessarily affect normal cells, including anti-cancer immune cells. Therefore, new therapeutic strategies devoted to the modulation of mitochondrial functions in a distinct cellular target need to be defined.

### 9.2. Intracellular Alkalinity of Cancer Cells

Recently, the Warburg effect has been explained as an effect of the intracellular alkalinity of cancer cells [207]. It was found that the glycolytic activation is itself an oncogenic event: indeed, the overexpression of the glucose transporter GLUT3 in nonmalignant human breast cells suffices to activate oncogenic signaling pathways such as EGFR, β1 integrin, MEK, and AKT, with consequent loss of tissue polarity and increased growth [207]. A proton [H^+^]-related mechanism involved in the initiation and progression of cancer has been recently described [208,209]. Specifically, it was shown that, in tumor cells, an extracellular acid microenvironment of 6.2–6.9 vs 7.3–7.4 in normal cells is associated with alkaline intracellular pH values of 7.12–7.7 vs 6.99–7.05 in normal cells [210,211,212]. This situation is linked to an elevated activity of the membrane-bound Na^+^/H^+^ exchanger isoform 1 (NHE1) [213,214]. Therefore, the use of NHE1 inhibitors with low side-effects such as Cariporide and other more potent NHE1 inhibitors, including compound 9t, have been recently proposed for cancer therapy [214].

### 9.3. Glutaminolysis and Cancer Therapy

Glutamine has some important functions in tumor metabolism and cancer cells depend on glutamine metabolism. Therefore, targeting glutamine metabolism could contribute to cancer therapy. Allosteric inhibitors of GLS (glutaminase) have shown promising results in cancer models. One of these inhibitors, BAPTES [bis-2-(5-phenylacetamido-1,2,4-thiadiazol-2-yl) ethyl sulfide] [215,216], inhibits the proliferation of cancer cells in vitro, and of xenografts in vivo, and prolongs survival in genetically engineered mouse models of cancer [217,218]. In addition, BAPTES increases radiation sensitivity of lung tumor cells and human lung tumor xenografts in mice [219]. Preclinical research has shown that CB-839, a GLS inhibitor, is efficacious against breast cancer and hematological malignancies [220,221]. 

GLS inhibition has been also found to be efficient in combination therapy. The inhibition of the anti-apoptotic protein BCL-2 integrates glutaminase inhibition [222]. Furthermore, it has been shown that highly invasive ovarian cancer cells have increased glutamine dependence compared to less invasive cells [223], and metastatic prostate tumors show increased dependence on glutamine uptake [223]. Furthermore, increased lactate in tumor microenvironment may promote increased glutamine metabolism though HIF2 and MYC-dependent mechanism [224], and increased ammonia released from cancer cells stimulates autophagy in the fibroblasts that consequently release additional glutamine, which is metabolized by the cancer cells [225] but may be toxic to surrounding cells. Interestingly, the lactate produced in cancer cells by increased glycolysis, which may be present in the tumor microenvironment, promotes glutamine metabolism by a HIF2 and MYC-dependent mechanism [226,227]. Increased glutaminolysis produces ammonia with consequent autophagy of exposed cells [228,229], such as fibroblasts, and further release of glutamine, which may be metabolized by cancer cells [230]. It must be also noted that some tumors, including those of brain and lung [231,232,233,234] synthesize and excrete glutamine. Ammonia may be toxic in this context and is probably detoxified by surrounding non-transformed cells. 

Glutaminolysis inhibition could be curative for highly invasive ovarian cancer cells exhibiting increased glutamine dependence compared to less invasive cancers [235]. Similarly, metastatic prostate tumors show increased dependence on glutamine uptake [223,224,236]. Furthermore, the genetic inhibition of glutaminase was shown to prevent epithelial-to-mesenchymal transition, a key step in tumor cell invasiveness and eventual metastasizing [237].

### 9.4. Nampt Inhibitors

Nampt has been considered a potential therapeutic target for cancer treatment, due to its contribution to cancer pathogenesis [238]. Nampt converts NAM (nicotinamide) and PRPP (5-phosphoribose-1-pyrophosphate) to nicotinamide mononucleotide, a substrate of Nmnat (nicotinamide mononucleotide adenylyl transferase) to generate NAD (nicotinamide adenine dinucleotide) by transferring the adenylyl moiety (Figure 8). An alternative way to generate NAD is the transformation of nicotinic acid (NA) to NAMN (nicotinic acid mononucleotide), catalyzed by Naprt (nicotinic acid phosphoribosyltransferase), followed by the synthesis of NAAD (nicotinic acid adenine dinucleotide), in presence of ATP, catalyzed by Nmnat (nicotinamide mononucleotide adenylyltransferase) and the transformation of NAAD to NA, catalyzed by a synthetase (Figure 8). Different Nampt inhibitors have been developed to date, among them FK866 (APO866) [239] reduces cellular levels of NAD and GAPDH [240,241], inhibits glycolysis and cancer cell growth [242] and induces apoptosis [239]. Another competitive Nampt inhibitor, GMX1778, has anti-cancer effects [243]. A recent study demonstrated that recurrent NAMPT-H191R mutations confer resistance to treatment of cancer cell lines with TF-31, but STF-31 also exerts an additional inhibitory effect against Nampt [244]. Although Nampt inhibition may prevent tumor cell growth, it is essential for normal cells and its absence may cause embryonic lethality, progressive muscle degeneration [245,246] and severe vision loss [247]. It must be noted, however, that Naprt (nicotinate phosphoribosyltransferase) generates nicotinic acid mononucleotide (NAMN) from nicotinic acid and PRPP (5-phosphoribose-1-pyrophosphate), and then Nmnat (nicotinamide mononucleotide adenylyltransferase) conjugates ATP to NAMN to generate NAAD (nicotinic acid adenine dinucleotide) (Figure 8) [248]. IDH (isocitrate dehydrogenase) mutations in glioma cell lines induce a decrease of Naprt expression via an increase of DNA histone methylations [249]. IDH mutations are frequent in patients with gliomas and acute myeloid leukemia (AML) [250]. Mutated IDH aberrantly generates 2-hydroxyglutarate that inhibits DNA and histone demethylases and promotes the hypermethylation of DNA histones [250]. 

## 10. Conclusions

In 1861, Louis Pasteur observed that oxygen increased the division of yeast and the inhibition of the fermentation by oxygen. In 1913 the mitochondria were discovered, and Otto Warburg linked cellular respiration to these organelles, derived from guinea pig liver extracts, that he called “grana” [251]. Warburg observed that tumors acquire the unusual property of fermenting glucose to lactate in the presence of oxygen (aerobic glycolysis), and proposed that deficient mitochondrial respiration causes aerobic glycolysis in cancer cells [251]. However, we now know that tumor mitochondria are well coupled (Figure 8) [7,89,138] and the Pasteur defect is largely linked to the great increase in the activity of glycolytic enzymes in cancer cells.

The fermentation of glucose to lactate in the presence of oxygen (Warburg effect) has been strongly criticized by Weinhouse [252] who, in opposition to the Warburg effect, observed that the well differentiated Morris hepatomas do not produce lactic acid in aerobiosis. Indeed, an almost complete absence of lactic acid production in four well-differentiated tumors has been found, and in aerobiosis, relatively low production in three well differentiated tumors, and high production in poorly-differentiated tumors (Figure 9). Altogether, these observations indicate that the Warburg effect is not the cause of the malignant transformation but its consequence, and an adaptation to hypoxia of tumor cells that becomes more evident during cancer progression (see the paragraph 3).

The first report on the Warburg effect was published 97 years ago [1,2]. Thirty-four years later the primary metabolic block produced by 2-DG [253] was localized, and forty years after the Warburg report of 1923, the inhibition of aerobic glycolysis by 2-DG [171] was demonstrated, as an inhibitor of HK and GPI [253], which inhibited protein synthesis in cancer cells. Some important recent studies reevaluate the old hypotheses. A series of elegant experiments has unequivocally shown that glycolytic metabolism activates the YAP/TAZ signal in cells of different neoplasms (e.g., breast and liver) and YAP activation upregulates the expression and transcriptional activity of HK2 and PFKFB3 [204,205]. Through the use of glycolysis inhibitors, it was observed that the YAP/TAZ signal, active in cells that incorporate glucose and produce lactic acid, is strongly inhibited when glucose metabolism is blocked, or glycolysis is reduced. It has been shown that the glycolytic enzyme PFK1, upregulated in tumors, promotes the YAP/TAZ transcriptional cooperation with TEAD factors to form the PFK1-TEAD1-YAP/TAZ complex in the nucleus. YAP activation during glycolysis, mediated by the hexosamine biosynthesis pathway, and its acetylation by O-linked b-N-acetylglucosamine transferase promote its nuclear translocation and transcriptional activity [182,254,255]. YAP/TEAD in the cell nucleus regulates GLUT3 transcription and activates YAP, thus augmenting HK2 expression and promoting glycolysis [182,255]. Therefore, the glycolytic metabolism regulates the transcription of YAP/TAZ [169] that promotes glycolysis, lipogenesis, and glutaminolysis, and is involved in the proliferative activity and aggressivity of neoplastic cells [182,255]. Collectively, this body of evidence clearly indicates that the Warburg effect is an epiphenomenon of the transformation process essential for the development of malignancy.

## Figures and Tables

**Figure 1 cancers-12-02819-f001:**
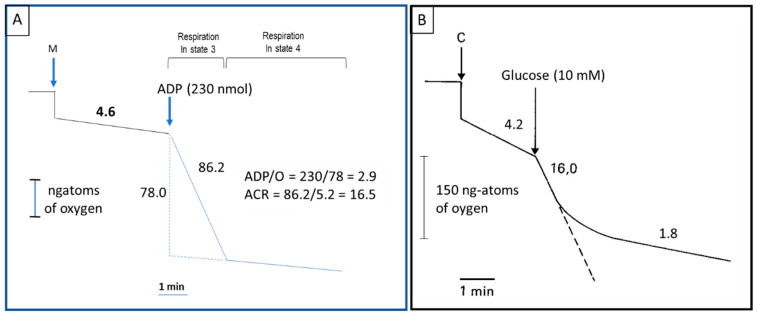
Respiratory control (**A**) and oxidative phosphorylation (**B**) of the mitochondria isolated from hepatocarcinoma ascites AH130. ACR, acceptor control ratio. C, cells, M, mitochondria.

**Figure 2 cancers-12-02819-f002:**
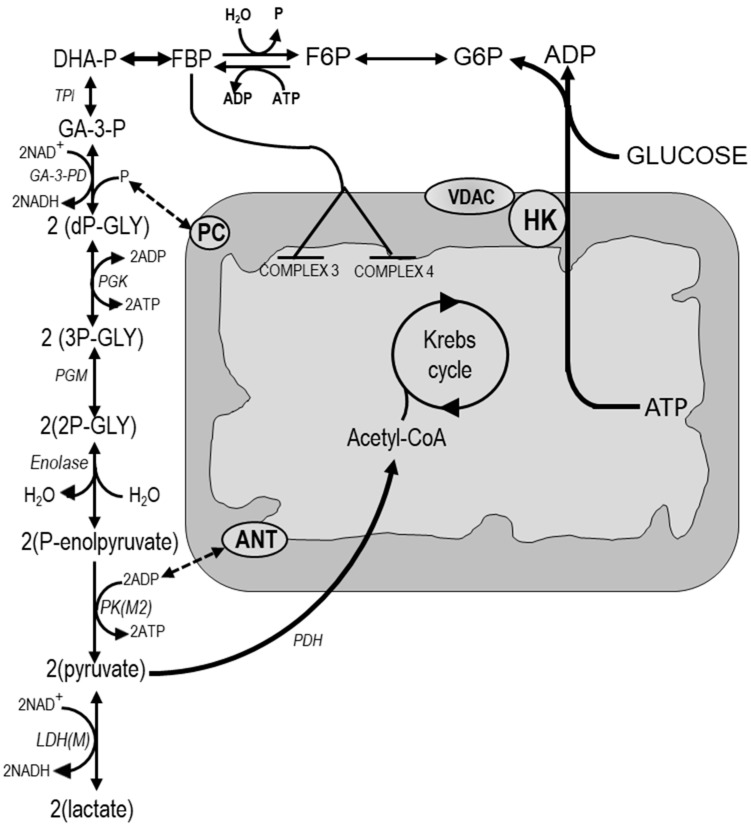
Reciprocal regulation between mitochondria and glycolytic pathway. Abbreviations: ANT, adenosin nucleotide transporter; PC, Pi carrier; VDAC, voltage-dependent anion channel. Substrates: DHA-P, dihydroxyacetone phosphate; F6P, fructose-6-phosphate; FBP, fructose-1,6-biphosphate; G6P, glucose-6-phosphate; 2dP-GLY, 2-diphosphoglycerate; 2PG, 2-phosphoglycerate; 3PG, 3-phosphoglycerate; PEP, phosphoenolpyruvate. Enzymes: GA-3-PD, glyceraldehyde-3-phosphate dehydrogenase; LDH, lactate dehydrogenase; PDH, pyruvate dehydrogenase; PGK, phosphoglycerate kinase; PGM, phosphoglycerate mutase; PKM2, pyruvate kinase M2; TPI, triose-phosphate isomerase. Dotted double arrows indicate competition.

**Figure 3 cancers-12-02819-f003:**
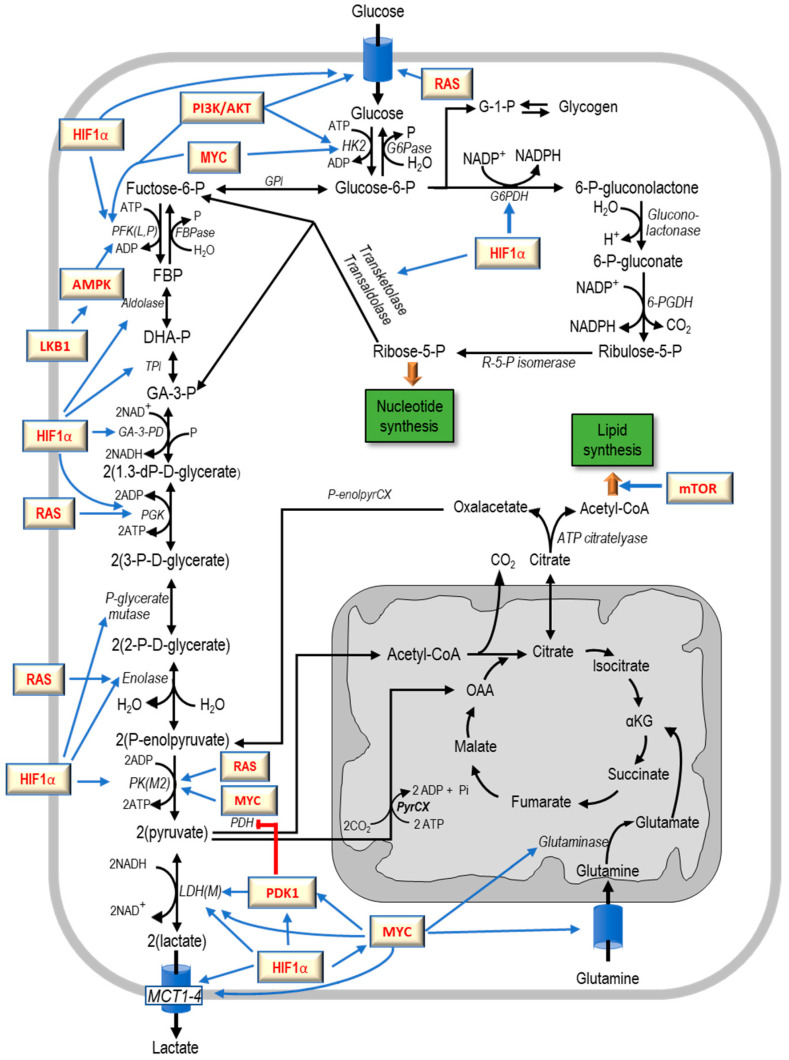
Glucose metabolism. Substrates: DHA-P, dihydroxyacetone phosphate; F6P, fructose-6-phosphate; FBP, fructose-1,6-biphosphate; G-1-P, glucose-1-phosphate; 2dP-GLY, 2-diphosphoglycerate; OAA, oxaloacetate; αKG, alpha-ketoglutarate; 2PG, 2-phosphoglycerate; 3PG, 3-phosphoglycerate; P-enolpyruvate, phosphoenolpyruvate. Enzymes: FBPase, fructosebiphosphatase; GA-3 PD, glyceraldehyde-3-phosphate dehydrogenase; GPI, glucose-phosphate isomerase; LDH, lactate dehydrogenase; PDH, pyruvate dehydrogenase; PGK, phosphoglycerate kinase; PGM, phosphoglycerate mutase; PKM2, (pyuvate kinase M2; PYRCX, pyruvate carboxylase; TPI, triose-phosphate isomerase. Genes are shown in the yellow rectangles. Blue arrows: activation; red blunt arrow: inhibition.

**Figure 4 cancers-12-02819-f004:**
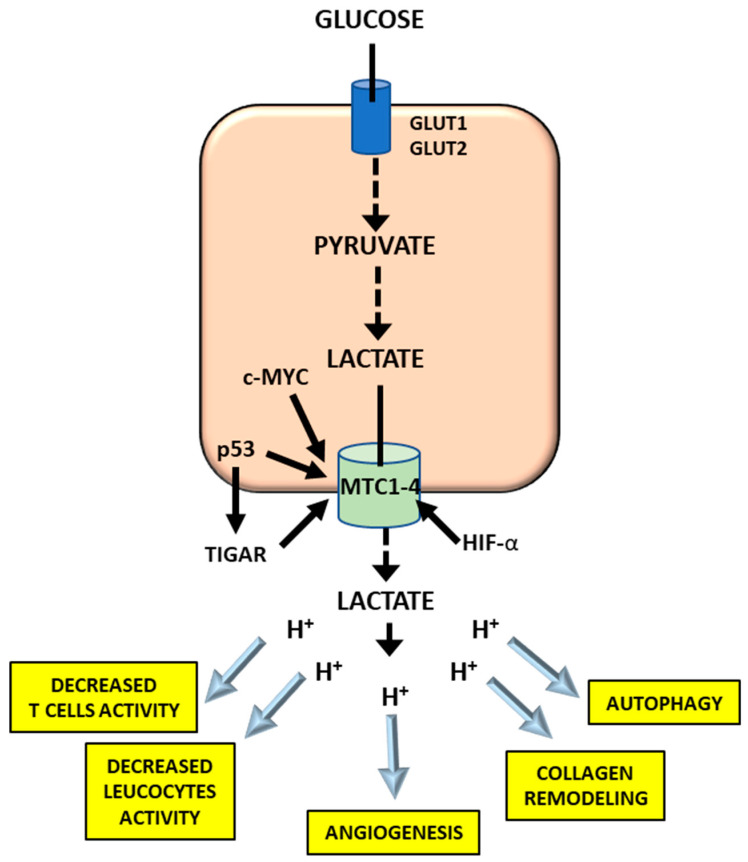
Effects of the lactic acid export of cells outside. Explanations in the text.

**Figure 5 cancers-12-02819-f005:**
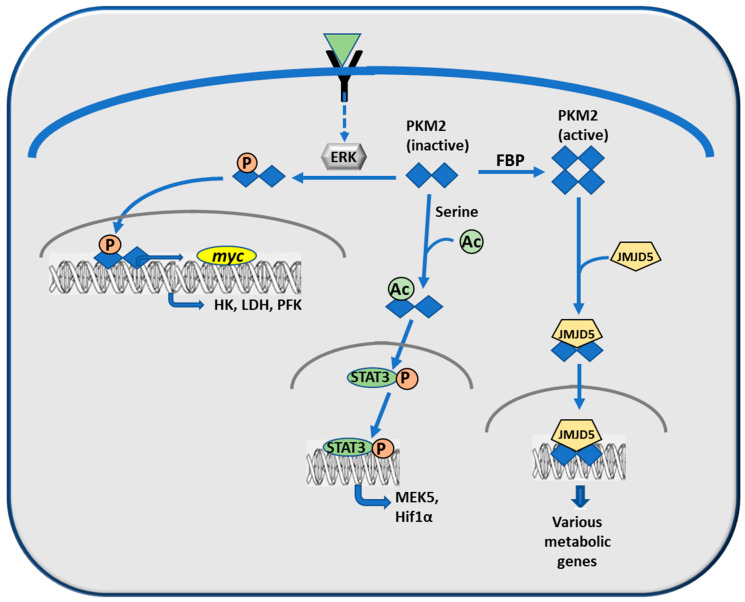
Modifications of dimeric and tetrameric PKM2 and nuclear effects of the different forms. Explanations in the text.

**Figure 6 cancers-12-02819-f006:**
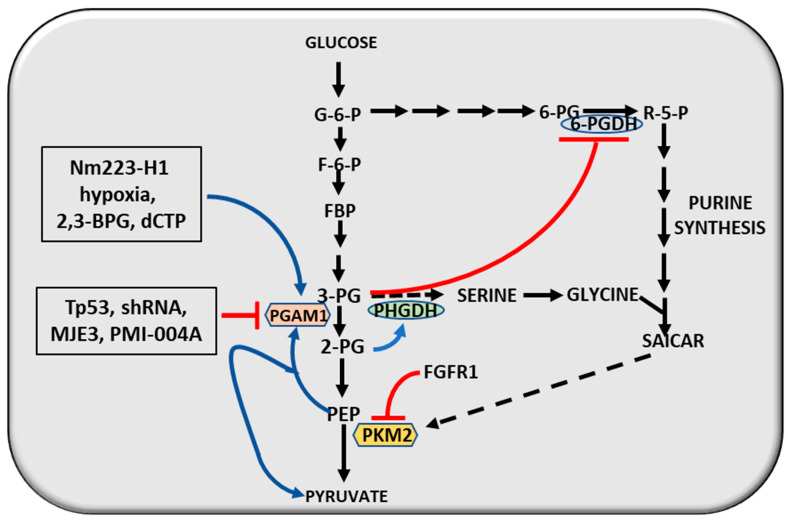
Reciprocal regulation of glycolysis and pentose phosphate pathway. Explanations in the text.

**Figure 7 cancers-12-02819-f007:**
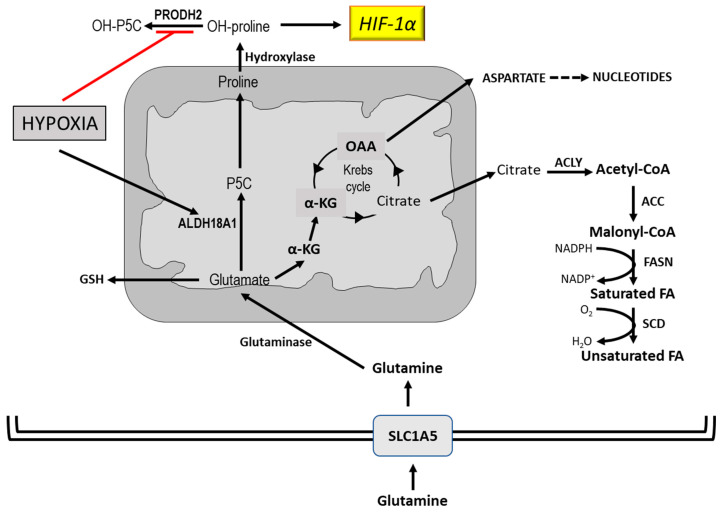
Role of glutamine in tumor metabolism and the support of HIF-dependent hypoxic response. Abbreviations: ACLY, ATP-dependent citrate lyase; ACC, Acetyl-CoA carboxylase; ALDH18A1, Aldehyde dehydrogenase 18 family, member a1; FASN, Fatty acids synthetase; P5C, proline-carboxylate; PRODH2, Hydroxyproline dehydrogenase; SCD, stearoyl CoA desaturase; SLC1A5, glutamine transporter.

**Figure 8 cancers-12-02819-f008:**
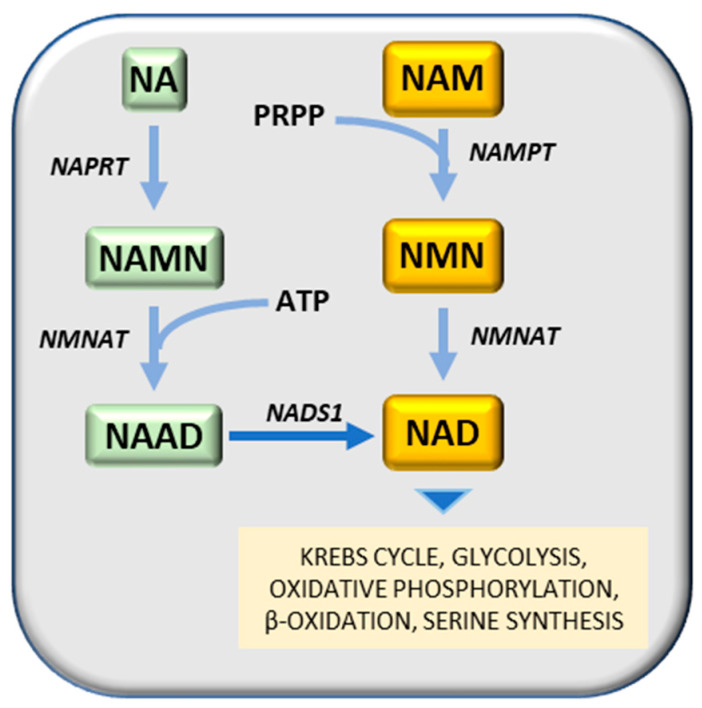
NAD synthesis. Substrates: NA, nicotinic acid; NAD, Nicotinamide adenine dinucleotide; NAAD, nicotinic acid adenine dinucleotide; NAM, nicotinamide; NAMN, nicotinic acid mononucleotide; NMN, nicotinamide mononucleotide; PRFPP, 5-phosphoribose-1-pyrophosphate. Enzymes: NADS1, NAD synthetase; NAMPT, nicotinamide phosphoribosyltransferase; NMNAT, nicotinamide mononucleotide adenylyltransferase; NAPRT, nicotinic acid phosphoribosyltransferase.

**Figure 9 cancers-12-02819-f009:**
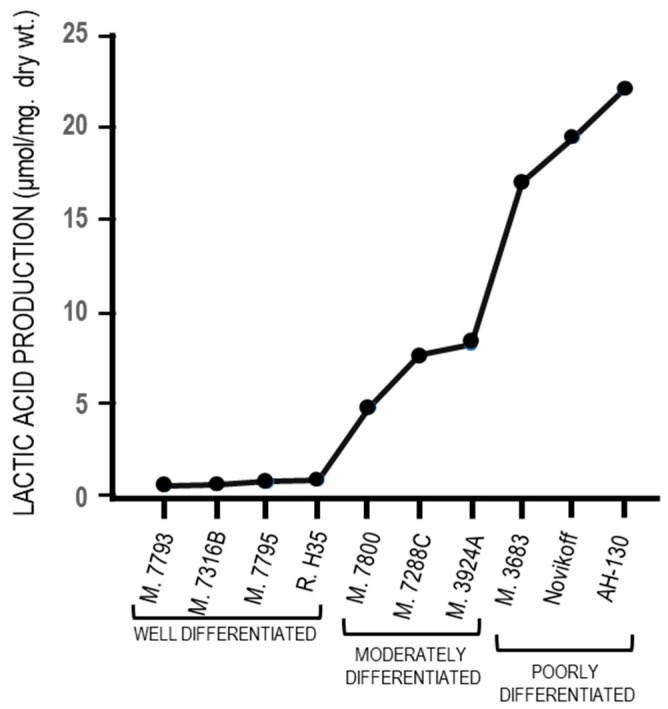
Aerobic glycolysis in hepatocellular carcinomas with different levels of differentiation.

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
