# Peer review of "The Warburg Effect 97 Years after Its Discovery"

_cancers, 2020, doi:10.3390/cancers12102819_

Round 1
Reviewer 1 Report
I consider all comments sufficiently edited and recommend that the manuscript be accepted for publication.
Reviewer 2 Report
Authors now addressed most of the reviewers' comments what increased a quality of this review article.
This manuscript is a resubmission of an earlier submission. The following is a list of the peer review reports and author responses from that submission.
Round 1
Reviewer 1 Report
The review article by Feo and colleagues on the Warburg Effect provides an excellent overview of this tumor-biologically important topic. The various issues are clearly structured and presented in a well-founded manner. The structuring in biochemical basics and involved factors, genetics, molecular biology, metabolism and mitochondria and finally therapeutic implications is very clear and guides well through the subject. The review article is very good and very important and should therefore be published in Cancers. However, two limitations of the manuscript need to be urgently addressed before it can be published.
Writing style and grammar are not up to the usual scientific standard. Sentences like 'This Phenomenon...' (lines 69-73) are difficult to read and not much easier to understand. I would therefore strongly recommend a revision by a native speaker, maybe you should also consider the use of a scientific correction service. The manuscript cannot be published in the present language supplement.
Furthermore, I would suggest that Figure 1 be revised. This figure does not meet the qualitative criteria for the publication of such an important review article in Cancers. I would suggest that an figure style similar to Figure 8 be used.
Reviewer 2 Report
I have major concerns which are listed below:
- This review is very descriptive without providing novel information on this topic. For example: what is the meaning and purpose of the Warburg effect; why do the cancer cells prefer glycolysis over oxidative respiration, etc.
- Isn't Warburg effect proliferation adaptation of cancer cells rather than a hallmark of cancer?
- Warburg's early citations - his discovery of cytochrome c oxidase (1920-1945) have to come up very early in the review, not at the end as authors did (references #202, 203). Reference #1 (Warburg, 1956) is not the original publication by Warburg.
- There are two #1 references in the References section. Also Warburg's 1956 reference appearch twice (#1 and #204).
- Authors should include in much more details "hot" topics that might provide insights for novel potential targets for cancer therapy such as glutamine metabolism (see Nguyen, 2018), NAD metabolism (see Yaku, 2018), hexosamine pathway, metabolic redox circuits (see Wang, 2019), etc.
- There is lots of literature reviews published on this topic (see Lee, 2015) so this review has to be significantly improved providing the comprehensive research observations.